# *Pestalotiopsis jiangsuensis* sp. nov. Causing Needle Blight on *Pinus massoniana* in China

**DOI:** 10.3390/jof10030230

**Published:** 2024-03-21

**Authors:** Hui Li, Bing-Yao Peng, Jun-Ya Xie, Yu-Qing Bai, De-Wei Li, Li-Hua Zhu

**Affiliations:** 1College of Forestry and Grassland, Nanjing Forestry University, Nanjing 210037, China; lhui@njfu.edu.cn (H.L.); nanlinpby@njfu.edu.cn (B.-Y.P.); xiejunya@njfu.edu.cn (J.-Y.X.); baiyuqing@njfu.edu.cn (Y.-Q.B.); 2Co-Innovation Center for Sustainable Forestry in Southern China, Nanjing Forestry University, Nanjing 210037, China; 3The Connecticut Agricultural Experiment Station Valley Laboratory, Windsor, CT 06095, USA; dewei.li@ct.gov

**Keywords:** multi-locus phylogeny, new species, pine

## Abstract

*Pinus massoniana* Lamb. is an important, common afforestation and timber tree species in China. Species of *Pestalotiopsis* are well-known pathogens of needle blight. In this study, the five representative strains were isolated from needle blight from needles of *Pi. massoniana* in Nanjing, Jiangsu, China. Based on multi-locus phylogenetic analyses of the three genomic loci (ITS, *TEF1*, and *TUB2*), in conjunction with morphological characteristics, a new species, namely *Pestalotiopsis jiangsuensis* sp. nov., was described and reported. Pathogenicity tests revealed that the five representative strains of the species described above were pathogenic to *Pi. massoniana*. The study revealed the diversity of pathogenic species of needle blight on *Pi. massoniana*. This is the first report of needle blight caused by *P. jiangsuensis* on *Pi. massoniana* in China and worldwide. This provides useful information for future research on management strategies of this disease.

## 1. Introduction

*Pinus massoniana* Lamb. is the most widely distributed timber tree species with the largest afforestation area in China [1], which provides a large amount of timber, oleoresin [2], carbon storage [3], and ecological products [4], and also has potential biomedical properties [5]. However, *Pi. massoniana* was found dead at the top of needles in plantations in Nanjing, Jiangsu Province with a high incidence, which seriously threatened the economic and ecological value.

Many pathogens have been reported to damage *Pi. massoniana* in the world; for example, its forestry and pine forests were threatened by outbreaks of pine wilt disease (PWD) caused by *Bursaphelenchus xylophilus* (pinewood nematode; PWN) [6]. Damping-off and root rot disease caused by *Fusarium oxysporum* has been found in *Pi. massoniana* [7,8]. *Pseudofusicoccum kimberleyense* and *Pse. violaceum* can cause dead branch disease of *Pi. massoniana* [9]. *Pestalotiopsis funerea* affected the needles of young *Pi. massoniana* trees and caused them to gradually dry up and fall off [10]. In addition, insect–parasitic entomopathogenic fungi such as *Penicillium citrinum*, *Purpurecillium lilacinum,* and *Fusarium* spp. were also confirmed to be pathogenic to *Pi. massoniana* [11]. However, as an important economic tree species, the host–pathogen relationship of *Pi. massoniana* needs more studies, and additional pathogens may be found.

*Pestalotiopsis* species are widely distributed in the world as endophytes, plant pathogens, or saprobes [12,13,14,15,16,17], mainly in tropical and temperate regions and have a wide range of host plants [15,18,19]. Initially, the characteristics of conidia, such as color, size, and appendages, are the key to the identification of *Pestalotiopsis* and related genera [20,21]. Those taxonomic groups related to the genus *Pestalotiopsis* are also called *pestalotioid* fungi. Afterwards according to the relationship between conidial morphology and multi-locus phylogeny [14,19,22,23], *Pestalotiopsis* sensu lato was divided into three genera by Maharachchikumbura et al. (2014) [15]—*Pestalotiopsis* sensu stricto, *Neopestalotiopsis,* and *Pseudopestalotiopsis*. Three genera correspond to three types of conidia, conidia with light brown or olivaceous concolorous median cells (*Pestalotiopsis* sensu stricto), conidia with versicolorous median cells (*Neopestalotiopsis*), and conidia with dark-colored concolorous median cells (*Pseudopestalotiopsis*) [14,19,22,24]. *Pestalotioid* species identification remains a major challenge because of the conidia of overlap, and the classification is complex [22,25,26].

Needle blight caused by *Pestalotiopsis* is a common disease in young pine forests, and the disease is widely distributed and causes serious damage. For example, *Pestalotiopsis funerea* can infect *Pinus tabulaeformis* [27], *Pi. taeda* [28], *Pi. massoniana* [10], etc. and cause needle blight. Xu et al. (2017) [29] reported that the pathogen causing the needle blight of *Pi. sylvestris* was *P. citrina*. The disease began to occur in 1974 and became popular in 1980, and it has become the main coniferous disease of trees [30,31]. Needle blight not only reduced the stock of trees but even led to the death of trees, which greatly threatened forestry production [32,33,34].

In March 2023, the needles of *Pi. massoniana* with the characteristics of needle blight were collected in Nanjing, Jiangsu Province. The earlier identification of *Pi. massoniana* needle blight in a previous study was in a different geographical area [10]; thus, the main purpose of this study was to determine the pathogen of *Pi. massoniana* needle blight and its pathogenicity by Koch’s postulates.

## 2. Materials and Methods

### 2.1. Field Survey and Fungal Isolation

In March 2023, needle lesions were found on *Pinus massoniana* in Lishui District, Nanjing, Jiangsu, China. The entire planting area of the *Pi. massoniana* forest was about 1800 m^2^. The symptoms of trees were visually observed and the needles with the symptoms were collected. Five symptomatic *Pi. massoniana* trees were randomly sampled. After macroscopic and microscopic observation of the collected pine needles, the pine needle fragments at the intermediate area of the diseased and healthy portions were cut off, and the surface was disinfected in 70% ethanol for 30 s, in 1% NaClO for 90 s, and then washed in sterile water for 90 s three times. Pine needle fragments were dried on sterile filter paper and incubated on potato dextrose agar (PDA) in the dark at 25 °C for 3 days. The hyphal tips of fungi emerging from tissue pieces were transferred to new PDA to obtain pure cultures. The isolates were obtained from needle blight samples of *Pi. massoniana*.

### 2.2. Morphological Identification

Colony morphology and pigment production on PDA was observed after 7 days at 25 °C with a 12/12 h light/dark cycle and inspected daily for fungal sporulation. Acervuli and conidial masses were observed under a Zeiss stereo microscope (SteRo Discovery v20, Oberkochen, Germany). The micromorphological characteristics of five isolates were observed by Zeiss Axio Imager A2m microscope (Carl Zeiss, Oberkochen, Germany), such as shape, color, septation, appendages, and size of conidia, conidiophores, and acervuli.

### 2.3. Genomic DNA Extraction, PCR, and Sequencing

Fungal genomic DNA of fungi cultured on PDA for 5 days was extracted by the cetyltrimethylammonium bromide (CTAB) method, and three distinct DNA regions were amplified by polymerase chain reactions (PCR). Three genomic loci, including the internal transcribed spacer (ITS), the partial translation elongation factor 1-alpha (*TEF1*), and partial β-tubulin (*TUB2*), were amplified with primers ITS5/ITS4 [35], EF1-728F/EF1-986R [36], and T1/Bt-2b [37,38], respectively. The protocols for amplification are shown in Table 1. Each 50 μL PCR mixture consisted of 25 μL of Premix TaqTM (Takara Biomedical Technology Company Limited, Beijing, China), 19 μL of dd H_2_O, 2 μL of forwarding primer, 2 μL of reverse primer, and 2 μL of DNA template. PCR purification and sequencing were performed by Sangon Biotech (Shanghai) Co., Ltd. (Nanjing, China).

### 2.4. Phylogenetic Analyses

Sequences with similarity of the ITS sequences generated in the present study were searched with the BLAST program on GenBank (https://blast.ncbi.nlm.nih.gov/, accessed on 3 November 2023), and the reference sequences used in this study were obtained. Concatenated multi-locus data (ITS, *TEF1*, and *TUB2*) were used for phylogenetic analyses with maximum likelihood (ML) and Bayesian Inference (BI). *Neopestalotiopsis protearum* (CBS 114178) was designated as an outgroup. The DNA sequences were aligned with MAFFT ver. 7.313 [39] and adjusted with BioEdit ver. 7.0.9.0 [40]. Maximum likelihood (ML) analysis was conducted on the multi-locus alignments using IQtree ver. 1.6.8 [41] with the GTR + F + I + G4 replacement model and the bootstrap method with 1000 replications to assess clade stability. RA × ML bootstrap support values were set at ML ≥ 70. Bayesian inference was analyzed using MrBayes ver. 3.2.6 with the GTR + I + G + F model (2 parallel runs, 2,000,000 generations) according to Quaedvlieg et al. (2014) [42]. Bayesian posterior probability values were set at PP ≥ 0.90. The phylogenetic trees were created in Figtree ver. 1.4.4. (http://tree.bio.ed.ac.uk/software/figtree/, accessed on 2 December 2023).

### 2.5. Genealogical Concordance Phylogenetic Species Recognition Analyses

The phylogenetically related ambiguous species were analyzed using the Genealogical Concordance Phylogenetic Species Recognition (GCPSR) to determine the recombination level in closely related species by performing a pairwise homoplasy index (PHI) test according to the method described by Quaedvlieg et al. (2014) [42]. A PHI result below 0.05 (Φw < 0.05) indicated significant recombination in the dataset. The relationships between closely related species were visualized in splits graphs with the LogDet transformation and splits decomposition options.

### 2.6. Pathogenicity Test

In this study, 12 two-year-old healthy *Pi. massoniana* seedlings and the three isolates representing the highest isolation frequency of *Pestalotiopsis* species were selected to perform the pathogenicity tests: BM 1-1, BM 1-2, BM 1-3—*Pestalotiopsis jiangsuensis* sp. nov. The tested plants were taken from the GuDong Green Seedling Base in Hechi, Guangxi Province, China. Healthy needles of *Pi. massoniana* were injured with a sterile needle. One wound was made per pine needle and conidial suspension (10^6^ conidia·mL^−1^) was sprayed on the wounds. Three plants were inoculated with each isolate, and the control was treated with sterile water. Inoculated seedlings and control seedlings were placed in a tent (1.5 × 1.2 × 1.5 m) with a humidifier (300 mL/h) to maintain RH 70%. The tent was placed in a greenhouse at 25 ± 2 °C and observed continuously for 10 days. All experiments were conducted three times.

## 3. Results

### 3.1. Disease Symptoms and Fungal Isolation

In March 2023, the incidence of needle blight of *Pi. massoniana* found in Nanjing, Jiangsu Province was ca. 60%, and the needle disease incidence of each *Pi. massoniana* was as high as 80%. The early symptom was a small yellow lesion at the needle tip, which extended from the needle tip downwards, and the lesion turned gray; a dark brown band encircled the needle at the junction with the healthy part (Figure 1A–C). Eventually the lesion area expanded until all the needles were necrotic. Ninety *Pestalotiopsis* strains were isolated and determined, based on the colony morphologies on PDA and ITS sequence blasting, with an isolation frequency of 90% (90/100). Five representative isolates (BM 1-1, BM 1-2, BM 1-3, BM 1-4, and BM 1-5) were selected for further study and deposited at the China Forestry Culture Collection Center (CFCC).

### 3.2. Phylogenetic Analyses

The concatenated sequence dataset of ITS, TEF1, and TUB2 included the five representative isolates, 120 taxa, and one outgroup taxon (*Neopestalotiopsis protearum* CBS 114178) with a total of 1637 base pairs (1-554 for the *TEF1*, 555-1163 for ITS, and 1164-1637 for *TUB2*) including gaps were obtained. The hosts, locations, and GenBank accession numbers of *Pestalotiopsis* species used for phylogenetic analyses in this study were shown in Table 2. The tree topology of the phylogenetic tree of ML and BI systems was congruent, and the bootstrap support values of RA × ML greater than 70% and the Bayesian posterior probabilities greater than 0.90 were denoted at nodes. In the phylogenetic analyses, five isolates formed a separate clade (ML/BI = 100/1), which was clustered into a big branch with four ex-type strains with a significant support (ML/BI = 98/0.92: *Pestalotiopsis foliicola* CFCC 54440, *P. pinicola* KUMCC 19-0183, *P. suae* CGMCC 3.23546, and *P. rosea* MFLUCC 12-0258. Based on the three-locus phylogenetic analyses and morphology, five strains (BM 1-1, BM 1-2, BM 1-3, BM 1-4, and BM 1-5) were identified as a new species of *Pestalotiopsis* (Figure 2).

Importantly, the PHI test of new species shows that no significant recombination (Φw = 0.071) events were observed between *Pestalotiopsis* sp. (undescribed taxon) and phylogenetically related species *P. foliicola* CFCC 54440, *P. pinicola* KUMCC 19-0183, *P. suae* CGMCC 3.23546, and *P. rosea* MFLUCC 12-0258 (Figure 3).

### 3.3. Taxonomy

***Pestalotiopsis jiangsuensis*** Li-Hua Zhu, Hui Li, and D.W. Li, sp. nov. Figure 4

**Index Fungorum No:** IF 900494

**Etymology:** the epithet referring to the province where the holotype was collected.

**Description:** Sporadic black and gregarious conidiomata produced on PDA after 7 days under light at 25 °C, globose, semi-immersed, dark brown to black, up to 400 μm diam (Figure 4B); conidiophores indistinct and reduced to conidiogenous cells. Conidiogenous cells (4.5-) 7.0–12.8 (−15.3) × (2.4-) 3.3 –5.6 (−6.5) µm (11.4 ± 2.5 × 4.4 ± 0.9 µm, *n* = 30), hyaline, ampulliform or cylindrical, and sometimes slightly wide at the base (Figure 4C). Conidia phragmospores, (20.3-) 22.1–25.5 (−27.3) × (6.2-) 6.7–8.2 (−8.7) µm (23.4 ± 1.8 × 7.5 ± 0.5 µm, *n* = 30), fusoid, ellipsoid, straight to slightly curved, 4-septate (Figure 4D); basal cell hyaline, obconic, thin-walled, 3.5–5.9 μm long; three median cells (12.7-) 13.7–15.5 (−16.5) × (6.2-) 6.7–7.4 (−7.9) µm (14.2 ± 1.0 × 7.2 ± 0.5 µm, *n* = 30), doliiform, wall rugose, concolorous, brown, septa darker than the rest of the cell (second cell from the base 4.2–5.9 μm long; third cell 4.8–5.7 μm long; fourth cell 4.0–5.4 μm long); apical cell hyaline, smooth-walled, conic or trapezoid, tapering toward the apex, 2.6–4.4 μm long, with 1–4 tubular apical appendages (mostly 2 and very few 4), arising from the apical crest, unbranched, filiform, 8.7–23.4 μm long; basal appendage single, tubular, unbranched, centric, 1.4–6.3 μm long.

**Culture characteristics:** Colonies on PDA flat with sparse aerial mycelia on the surface after 7 d at 25 °C, edge undulate, pale honey-colored, and reverse pale brown in the center and pale luteous margin (Figure 4A).

**Holotype:** China, Jiangsu province, Nanjing city, Lishui district, Baima National Agricultural Science and Technology Park, 119°10′44″ N, 31°36′28″ E (DMS), isolated from needles of *Pinus massoniana*, 1 March 2023, Hui Li, holotype CFCC 59538. Holotype is a living specimen being maintained via lyophilization at the China Forestry Culture Collection Center (CFCC), Chinese Academy of Forestry, Beijing, China, and ex-type BM 1-1 is stored at Forest Pathology Laboratory, Nanjing Forestry University.

**Habitat and host:** On needles of *Pinus massoniana* with needle blight.

**Known distribution:** Nanjing, Jiangsu Province, China.

**Additional specimens examined:** China, Jiangsu province, Nanjing city, Lishui district, Baima National Agricultural Science and Technology Park, 119°10′44″ N, 31°36′28″ E (DMS), isolated from needles of *Pinus massoniana*, 1 March 2023, Hui Li, cultures: CFCC 59539 (=BM 1-2), CFCC 59540 (=BM 1-3), CFCC 59541 (=BM 1-4), and CFCC 59542 (=BM 1-5).

**Notes:** *Pestalotiopsis jiangsuensis* is a species often having one to four tubular apical appendages, which are phylogenetically and morphologically well distinguished from *P. foliicola* CFCC 54440, *P. pinicola* KUMCC 19-0183, *P. suae* CGMCC 3.23546, and *P. rosea* MFLUCC 12-0258. Although the five strains studied are a sister clade of *P. foliicola* CFCC 54440, *P. pinicola* KUMCC 19-0183, *P. suae* CGMCC 3.23546, and *P. rosea* MFLUCC 12-0258, the number of apical appendages is quite different. *Pestalotiopsis folicola*, *P. pinicola* and *P. suae* have two to three apical appendages; P. rosea has one to three tubular apical appendages, and some appendages are branched. The strains in this study have one to four apical appendages, and the appendages are unbranched.

*Pestalotiopsis funerea* has two to four apical appendages, and *Pestalotiopsis lawsoniae* has two apical appendages. They also have differences with *P. jiangsuensis*. In addition, *P. funerea* has a longer basal appendage than that of *P. jiangsuensis* (5–7) µm vs. (1.4–6.3) µm [43,44].

### 3.4. Pathogenicity Test

In the experiment of Koch’s postulates, the three representative isolates were pathogenic to *Pi. massoniana* needles. The development of disease symptoms was observed during a 10-day period. At 5 d, all the *Pestalotiopsis jiangsuensis* isolates developed gray to gray-brown lesions on wounded needles of *Pi. massoniana* (Figure 5B–D). At 10 d, the lesion expanded, and in severe cases, the whole needle was necrotic (Figure 5F–H). No symptoms developed on the needles of the control (Figure 5A,E). In this study, the pathogenicity of *Pestalotiopsis jiangsuensis* is strong; for example, the lesions spread almost to the whole needle after 10 days. It may also relate to its high isolation rate. *Pestalotiopsis jiangsuensis* was successfully re-isolated from 100% of the inoculated plants and identified based on morphological features and phylogenetic analysis of ITS. Thus, Koch’s postulates had been fulfilled.

## 4. Discussion

*Pestalotiopsis* was established by Steyeart (1949) [45] and typified with *Pestalotiopsis guepinii* Steyaert. *Pestalotiopsis* sensu lato was classified based on conidia with five-celled, the middle three intermediate colored cells, and hyaline end cells. After that, its taxonomic characteristics gradually changed into conidia spindle-shaped, with five-celled, with colorless or nearly colorless cells at both ends, dark cells in the middle, and one or more branched or unbranched apical appendages arising from the apical cell, with or without basal stalk [20,21,46,47]. The excessive overlap of conidia makes it difficult to identify *Pestalotioid* species only by morphological characteristics [19]. Although some additional taxonomic features can also be used as the basis for the identification of *Pestalotiopsis*—such as the pigmentation of median cells, which is an important character to distinguish *Pestalotiopsis funerea* and *P. triseta* [23,48]—there are still great limitations [17,22,49]. However, the application of molecular data in the identification of *Pestalotiopsis* species has greatly improved the accuracy and credibility [22,23,26,50,51]. *Pestalotiopsis* sensu lato was segregated into three genera by Maharachchikumbura et al. (2014) [15] as *Pestalotiopsis* sensu stricto, *Neopestalotiopsis*, and *Pseudopestalotiopsis*, based on both morphological characteristics and phylogenetic analyses. Gu et al. (2022) [17] identified six new *Pestalotiopsis* species from Rhododendron, based on phylogenetic analyses of combined ITS, *TEF1*, and *TUB2* genes/region along with morphological characteristics. Maharachchikumbura et al. (2012) [14] identified 23 species of *Pestalotiopsis* from different host plants in China, including 14 new species, based on phylogenetic analysis of ITS, *TEF1,* and *TUB2* genes/region and morphology. More importantly, concatenating ITS, *TUB2,* and *TEF1* sequences can provide better identification information for *Pestalotiopsis* [14,52].

The Global Biodiversity Information Facility (https://www.gbif.org/, accessed on 24 November 2023) displays 9320 records of *Pestalotiopsis* from all over the world, including years and coordinates [53]. The data show that most of them are distributed in Australia, Brazil, China, and the United States. *Pestalotiopsis* as a plant pathogen has a wide range of symptoms on the hosts, such as withering or chlorosis of leaves, dead shoots or tips, and canker [15]. In *Pinus* spp., it may be characterized by shoot blight, trunk necrosis, needle blight, and pinecone decay [54]. It is not uncommon that a species of *Pestalotiopsis* was successfully isolated from needles of *Pinus* species [34]. For example, *Pestalotiopsis neglecta* and *P. citrina* isolated from *Pi. sylvestris* can cause the needles to turn yellow partially or completely and even cause death of the trees [29,34]. *Pestalotiopsis bessey* isolated from *Pi. halenpesis* can cause the entire needles to turn dark gray-brown and eventually cause the death of the trees [55,56]. *Pestalotiopsis pini* isolated from *Pi. Pinea* can cause the needles and branches to wither, trunk necrosis, and pinecone rot [54]. *Pestalotiopsis* is also an endophytic fungus of some *Pinus* spp., such as *P. funerea*, and it was isolated from the healthy needles of *Pi. pinaster* [57].

Interestingly, the pathogen of *Pi. massoniana* needle blight isolated in a previous study was *P. funerea* [58], but the pathogen obtained in this study was *Pestalotiopsis jiangsuensis*, which indicated that the pathogens of the same genus on the same host were diverse. Silva et al. [54] isolated *P. disseminata* and *P. pini* from *Pi. Pinea,* and their results also confirmed this view. Similarly, the same species of *Pestalotiopsis* can be found on different plant hosts, such as *P. funereal,* which was isolated from *Pi. tabulaeformis*, *Pi. taeda*, and *Pi. massoniana* [10,27,28]. *Pestalotiopsis chamaeropis* was isolated from *Quercus* sp., *Castanopsis* sp., and *Camellia* sp. [15,49,59]. However, in the current study the samples were only collected from one site. In future research, the investigation areas should be expanded to study fungal diversity on *Pinus* spp. and related ecological functions.

## 5. Conclusions

In this study, we examined five strains, all of which were pathogenic to *Pi. massoniana*. Combined with morphology, multi-locus phylogenetic analyses, and GCPSR principle, these five strains were identified to be a new species to science, *Pestalotiopsis jiangsuensis*. This is the first report of needle blight caused by *P. jiangsuensis* on *Pi. massoniana* in China and worldwide, and it will provide useful information for future studies on all the phytopathological perspectives of this fungus and the management strategies of this newly emerged disease.

## Figures and Tables

**Figure 1 jof-10-00230-f001:**
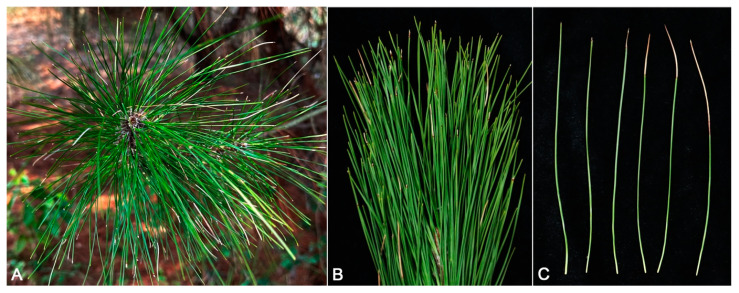
Symptoms of needle blight on *Pinus massoniana* in the field (**A**–**C**).

**Figure 2 jof-10-00230-f002:**
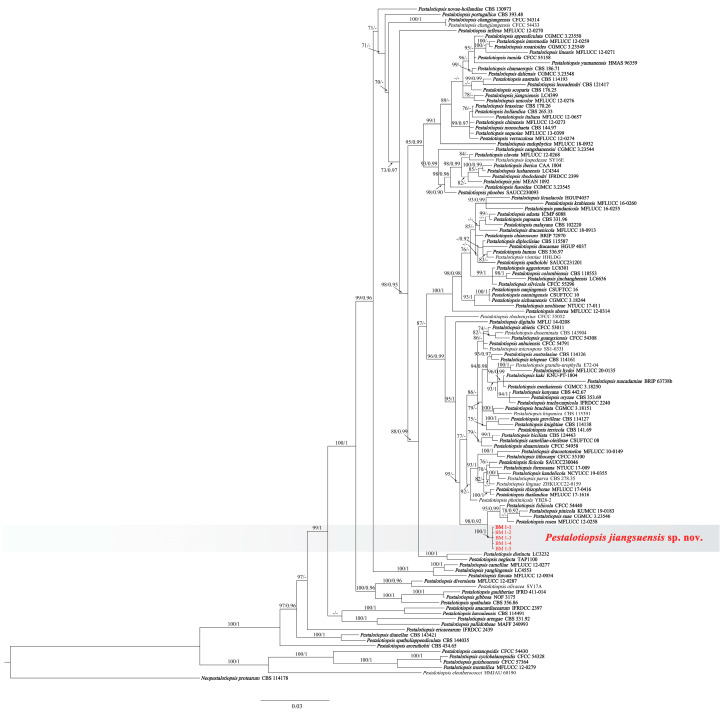
Phylogenetic relationship of *Pestalotiopsis jiangsuensis* isolates: BM 1-1, BM 1-2, BM 1-3, BM 1-4, and BM 1-5, based on concatenated sequences of ITS, TEF1, and TUB2 genes/region. RA × ML bootstrap support values (ML ≥ 70) and Bayesian posterior probability values (PP ≥ 0.90) were shown at the nodes (ML/PP). *Neopestalotiopsis protearum* (CBS 114178) is used as an outgroup. Bar = 0.04 substitution per nucleotide position. The sequences from this study are in red. The ex-type strains are in bold.

**Figure 3 jof-10-00230-f003:**
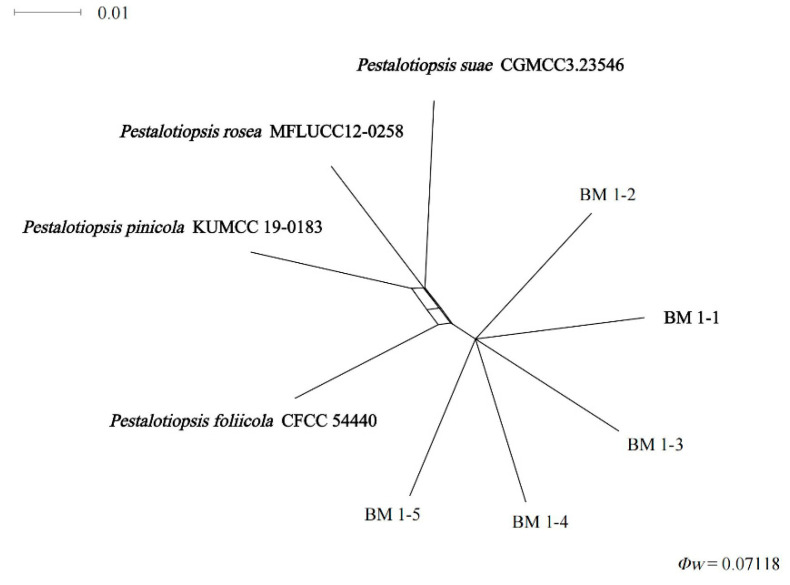
Pairwise homoplasy index (PHI) test of *Pestalotiopsis* isolates: BM 1-1, BM 1-2, BM 1-3, BM 1-4, and BM 1-5 and closely related *P. foliicola*, *P. pinicola*, *P. suae,* and *P. rosea* using both LogDet transformation and splits decomposition. PHI test results (Φw) < 0.05 indicate significant recombination within the data set.

**Figure 4 jof-10-00230-f004:**
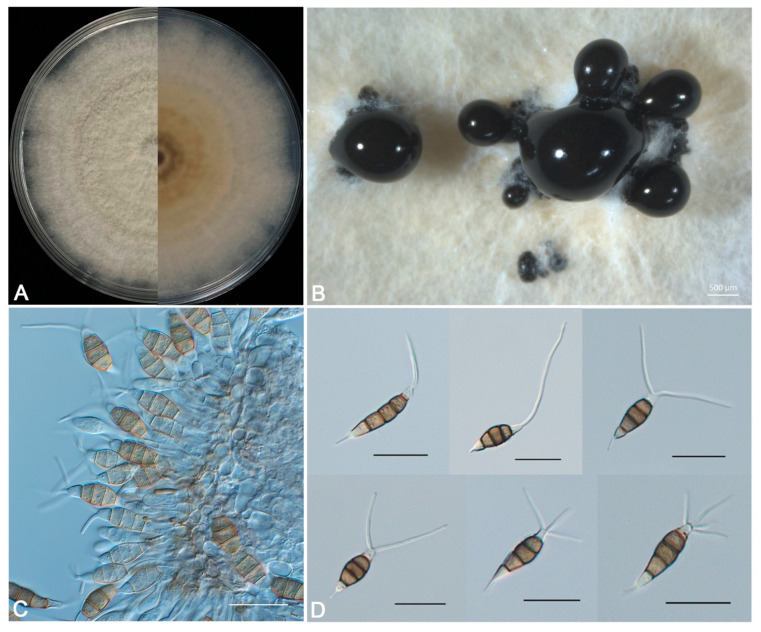
Morphological characteristics of *Pestalotiopsis jiangsuensis* sp. nov. BM 1-1. (**A**) Fungal colony on PDA, 5 d growth from above (L) and below (R). (**B**) Conidiomata and conidial masses. (**C**) Conidiophores, conidiogenous cells, and conidia. (**D**) Conidia. Scale bars: (**B**) = 500 μm, (**C**,**D**) = 20 μm.

**Figure 5 jof-10-00230-f005:**
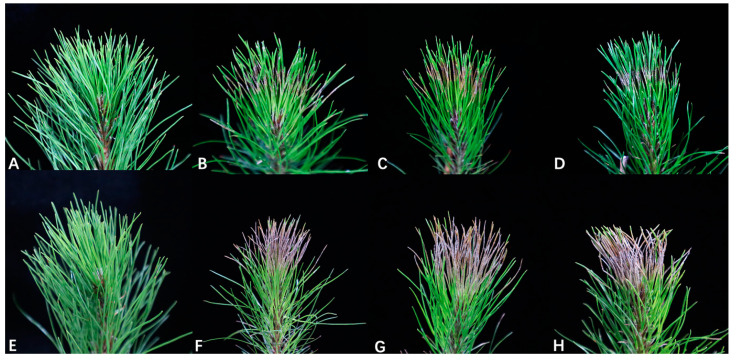
Pathogenicity of representative isolates of *Pestalotiopsis jiangsuensis* sp. nov. (BM 1-1, BM 1-2, and BM 1-3) on *Pinus massoniana*. (**A**) No symptoms were observed on control pine needles treated with sterile water after 5 days. (**B**–**D**) Symptoms on pine needles inoculated with conidial suspensions of BM 1-1, BM 1-2, and BM 1-3 after 5 days, respectively. (**E**) No symptoms observed on control pine needles treated with sterile water after 10 days. (**F**–**H**) Symptoms on pine needles inoculated with conidial suspensions of BM 1-1, BM 1-2, and BM 1-3 after 10 days.

**Table 1 jof-10-00230-t001:** Reaction conditions used in PCR amplification and sequencing.

Locus	PCR Primers (Forward/Reverse)	PCR: Thermal Cycles: (Annealing Temperature in Bold)
ITS	ITS5/ITS4	94 °C: 3 min, (94 °C: 45 s, **55 °C**: 45 s, 72 °C: 1 min) ×35 cycles, 72 °C: 10 min
*TEF1*	EF1-728F/EF1-986R	94 °C: 3 min, (94 °C: 45 s, **55 °C**: 45 s, 72 °C: 1 min) ×35 cycles, 72 °C: 10 min
*TUB2*	T1/Bt-2b	94 °C: 3 min, (94 °C: 45 s, **56 °C**: 60 s, 72 °C: 1 min) ×35 cycles, 72 °C: 10 min

**Table 2 jof-10-00230-t002:** Host, Origin, and GenBank accession numbers of strains of *Pestalotiopsis* species used for phylogenetic analyses.

Species ^a^	Strain Number ^b^	Host	Origin	GenBank Accession Number ^c^		
				ITS	*TUB2*	*TEF1*
*Pestalotiopsis abietis*	CFCC 53011 ^T^	*Abies fargesii*	China	MK397013	MK622280	MK622277
*P. adusta*	ICMP 6088 ^T^	*Prunus cerasus*	Fiji	JX399006	JX399037	JX399070
*P. aggestorum*	LC6301 ^T^	*Camellia sinensis*	China	KX895015	KX895348	KX895234
*P. anacardiacearum*	IFRDCC 2397 ^T^	*Mangifera indica*	China	KC247154	KC247155	KC247156
*P. anhuiensis*	CFCC 54791 ^T^	*Cyclobalanopsis glauca*	China	ON007028	ON005056	ON005045
*P. appendiculata*	CGMCC 3.23550 ^T^	*Rhododendron decorum*	China	OP082431	OP185516	OP185509
*P. arengae*	CBS 331.92 ^T^	*Arenga undulatifolia*	Singapore	KM199340	KM199426	KM199515
*P. arceuthobii*	CBS 434.65 ^T^	*Arceuthobium campylopodum*	USA	KM199341	KM199427	KM199516
*P. australasiae*	CBS 114126 ^T^	*Knightia* sp.	New Zealand	KM199297	KM199409	KM199499
*P. australis*	CBS 114193 ^T^	*Grevillea* sp.	Australia	KM199332	KM199383	KM199475
*P. biciliata*	CBS 124463 ^T^	*Platanus × hispanica*	Slovakia	KM199308	KM199399	KM199505
*P. brachiata*	CGMCC 3.18151 ^T^	*Rhizophora apiculata*	Thailand	MK764274	MK764340	MK764318
*P. brassicae*	CBS 170.26 ^T^	*Brassica napus*	New Zealand	KM199379	-	KM199558
*P. camelliae*	MFLUCC 12-0277 ^T^	*Camellia japonica*	China	JX399010	JX399041	JX399074
*P. camelliae-oleiferae*	CSUFTCC 08 ^T^	*Camellia oleifera*	China	OK493593	OK562368	OK507963
*P. cangshanensisi*	CGMCC 3.23544 ^T^	*Rhododendron delavayi*	China	OP082426	OP185517	OP185510
*P. castanopsidis*	CFCC 54430 ^T^	*Castanopsis lamontii*	China	OK339732	OK358508	OK358493
*P. chamaeropis*	CBS 186.71 ^T^	*Chamaerops humilis*	Italy	KM199326	KM199391	KM199473
*P. changjiangensis*	CFCC 54314 ^T^	*Castanopsis tonkinensis*	China	OK339739	OK358515	OK358500
*P. changjiangensis*	CFCC 54433	*Castanopsis tonkinensis*	China	OK339740	OK358516	OK358501
*P. chiaroscuro*	BRIP 72970 ^T^	*Sporobolus natalensis*	Australia	OK422510	-	-
*P. chinensis*	MFLUCC 12-0273 ^T^	*Taxus* sp.	China	JX398995	-	-
*P. clavata*	MFLUCC 12-0268 ^T^	*Buxus* sp.	China	JX398990	JX399025	JX399056
*P. colombiensis*	CBS 118553 ^T^	*Eucalyptus urograndis*	Colombia	KM199307	KM199421	KM199488
*P. cyclobalanopsidis*	CFCC 54328 ^T^	*Cyclobalanopsis glauca*	China	OK339735	OK358511	OK358496
*P. daliensis*	CGMCC 3.23548 ^T^	*Rhododendron decorum*	China	OP082429	OP185511	OP185518
*P. dianellae*	CBS 143421 ^T^	*Dianella* sp.	Australia	MG386051	MG386164	-
*P. digitalis*	MFLU 14-0208 ^T^	*Digitalis purpurea*	New Zealand	KP781879	KP781883	-
*P. diploclisiae*	CBS 115587 ^T^	*Diploclisia glaucescens*	China	KM199320	KM199419	KM199486
*P. disseminata*	CBS 143904	*Persea americana*	New Zealand	MH554152	MH554825	MH554587
*P. distincta*	LC3232 ^T^	*Camellia sinensis*	China	KX894961	KX895293	KX895178
*P. diversiseta*	MFLUCC12-0287 ^T^	*Rhododendron* sp.	China	JX399009	JX399040	JX399073
*P. dracaenae*	HGUP 4037 ^T^	*Dracaena fragrans*	China	MT596515	MT598645	MT598644
*P. dracaenicola*	MFLUCC 18-0913 ^T^	*Dracaena* sp.	Thailand	MN962731	MN962733	MN962732
*P. dracontomelon*	MFLUCC 10-0149 ^T^	*Dracontomelon dao*	Thailand	KP781877	-	KP781880
*P. eleutherococci*	HMJAU 60190	*Eleutherococcus brachypus*	China	OL996127	OL898722	-
*P. endophytica*	MFLUCC 18-0932 ^T^	*Magnolia garrettii*	Thailand	MW263946	-	MW417119
*P. ericacearum*	IFRDCC 2439 ^T^	*Rhododendron delavayi*	China	KC537807	KC537821	KC537814
*P. etonensis*	BRIP 66615 ^T^	*Sporobolus jacquemontii*	Australia	MK966339	MK977634	MK977635
*P. ficicola*	SAUCC230046 ^T^	*Ficus microcarpa*	China	OQ691974	OQ718749	OQ718691
*P. foliicola*	CFCC 54440 ^T^	*Castanopsis faberi*	China	ON007029	ON005057	ON005046
*P. formosana*	NTUCC 17-009 ^T^	*Neolitsea villosa*	China	MH809381	MH809385	MH809389
*P. furcata*	MFLUCC 12-0054 ^T^	*Camellia sinensis*	Thailand	JQ683724	JQ683708	JQ683740
*P. fusoidea*	CGMCC 3.23545 ^T^	*Rhododendron delavayi*	China	OP082427	OP185519	OP185512
*P. gaultheriae*	IFRD 411-014 ^T^	*Gaultheria forrestii*	China	KC537805	KC537819	KC537812
*P. gibbosa*	NOF 3175 ^T^	*Gaultheria shallon*	Canada	LC311589	LC311590	LC311591
*P. grandis-urophylla*	E72-04	*Eucalyptus grandis*	Brazil	KU926710	KU926718	KU926714
*P. grevilleae*	CBS 114127 ^T^	*Grevillea* sp.	Australia	KM199300	KM199407	KM199504
*P. guangxiensis*	CFCC 54308 ^T^	*Quercus griffithii*	China	OK339737	OK358513	OK358498
*P. guizhouensis*	CFCC 57364 ^T^	*Cyclobalanopsis glauca*	China	ON007035	ON005063	ON005052
*P. hawaiiensis*	CBS 114491 ^T^	*Leucospermum* sp.	USA	KM199339	KM199428	KM199514
*P. hispanica*	CBS 115391	*Eucalyptus globulus*	Portugal	MW794107	MW802840	MW805399
*P. hollandica*	CBS 265.33 ^T^	*Sciadopitys verticillata*	Netherlands	KM199328	KM199388	KM199481
*P. humus*	CBS 336.97 ^T^	Soil	Papua New Guinea	KM199317	KM199420	KM199484
*P. hydei*	MFLUCC 20-0135 ^T^	*Litsea petiolata*	Thailand	MW266063	MW251112	MW251113
*P. iberica*	CAA 1004 ^T^	*Pinus radiata*	Spain	MW732248	MW759035	MW759038
*P. inflexa*	MFLUCC 12-0270 ^T^	Unidentified tree	China	JX399008	JX399039	JX399072
*P. intermedia*	MFLUCC 12-0259 ^T^	Unidentified tree	China	JX398993	JX399028	JX399059
*P. italiana*	MFLUCC 12-0657 ^T^	*Cupressus glabra*	Italy	KP781878	KP781882	KP781881
** *P. jiangsuensis* **	**CFCC 59538**	** *Pinus massoniana* **	**China**	**OR533577**	**OR539191**	**OR539186**
	**CFCC 59539**			**OR533578**	**OR539192**	**OR539187**
	**CFCC 59540**			**OR533579**	**OR539193**	**OR539188**
	**CFCC 59541**			**OR533580**	**OR539194**	**OR539189**
	**CFCC 59542**			**OR533581**	**OR539195**	**OR539190**
*P. jiangxiensis*	LC4399 ^T^	*Camellia* sp.	China	KX895009	KX895341	KX895227
*P. jinchanghensis*	LC6636 ^T^	*Camellia sinensis*	China	KX895028	KX895361	KX895247
*P. kaki*	KNU-PT-1804 ^T^	*Diospyros kaki*	Korea	LC552953	LC552954	LC553555
*P. kandelicola*	NCYUCC 19-0355 ^T^	*Kandelia candel*	China	MT560723	MT563100	MT563102
*P. kenyana*	CBS 442.67 ^T^	*Coffea* sp.	Kenya	KM199302	KM199395	KM199502
*P. knightiae*	CBS 114138 ^T^	*Knightia* sp.	New Zealand	KM199310	KM199408	KM199497
*P. krabiensis*	MFLUCC 16-0260 ^T^	*Pandanus* sp.	Thailand	MH388360	MH412722	MH388395
*P. lespedezae*	SY16E	*Pinus armandii*	China	EF055205	-	EF055242
*P. leucadendri*	CBS 121417 ^T^	*Leucadendron* sp.	South Africa	MH553987	MH554654	MH554412
*P. licualacola*	HGUP4057 ^T^	*Licuala grandis*	China	KC492509	KC481683	KC481684
*P. linearis*	MFLUCC 12-0271 ^T^	*Trachelospermum* sp.	China	JX398992	JX399027	JX399058
*P. linguae*	ZHKUCC 22-0159	*Pyrrosia lingua*	China	OP094104	OP186108	OP186110
*P. lithocarpi*	CFCC 55100 ^T^	*Lithocarpus chiungchungensis*	China	OK339742	OK358518	OK358503
*P. lushanensis*	LC4344 ^T^	*Camelia* sp.	China	KX895005	KX895337	KX895223
*P. macadamiae*	BRIP 63738b ^T^	*Macadamia integrifolia*	Australia	KX186588	KX186680	KX186621
*P. malayana*	CBS 102220 ^T^	*Macaranga triloba*	Malaysia	KM199306	KM199411	KM199482
*P. menhaiensis*	CGMCC 3.18250 ^T^	*Camellia sinensis*	China	KU252272	KU252488	KU252401
*P. microspora*	SS1-033I	*Cornus canadensis*	Canada	MT644300	-	-
*P. monochaeta*	CBS 144.97 ^T^	*Quercus robur*	Netherlands	KM199327	KM199386	KM199479
*P. montellica*	MFLUCC12-0279 ^T^	*Fagraea bodeni*	China	JX399012	JX399043	JX399076
*P. nanjingensis*	CSUFTCC 16 ^T^	*Camellia oleifera*	China	OK493602	OK562377	OK507972
*P. nanningensis*	CSUFTCC 10 ^T^	*Camellia oleifera*	China	OK493596	OK562371	OK507966
*P. neglecta*	TAP1100 ^T^	*Quercus myrsinaefolia*	Japan	AB482220	LC311599	LC311600
*P. neolitseae*	NTUCC 17-011 ^T^	*Neolitsea villosa*	China	MH809383	MH809387	MH809391
*P. novae-hollandiae*	CBS 130973 ^T^	*Banksia grandis*	Australia	KM199337	KM199425	KM199511
*P. olivacea*	SY17A	*Pinus armandii*	China	EF055215	EF055251	-
*P. oryzae*	CBS 353.69 ^T^	*Oryza sativa*	Denmark	KM199299	KM199398	KM199496
*P. pallidotheae*	MAFF 240993 ^T^	*Pieris japonica*	Japan	AB482220	LC311584	LC311585
*P. pandanicola*	MFLUCC 16-0255 ^T^	*Pandanus* sp.	Thailand	MH388361	MH412723	MH388396
*P. papuana*	CBS 331.96 ^T^	Coastal soil Papua	New Guinea	KM199321	KM199413	KM199491
*P. parva*	CBS 278.35	*Leucothoe fontanesiana*	Thailand	KM199313	KM199405	KM199509
*P. phoebes*	SAUCC230093 ^T^	*Phoebe zhenna*	China	OQ692028	OQ718803	OQ718745
*P. photinicola*	YB28-2	Mango	China	MK228997	MK360938	MK512491
*P. pini*	MEAN 1092 ^T^	*Pinus pinea*	Portugal	MT374680	MT374705	MT374693
*P. pinicola*	KUMCC 19-0183 ^T^	*Pinus armandii*	China	MN412636	MN417507	MN417509
*P. portugallica*	CBS 393.48 ^T^	*-*	Portugal	KM199335	KM199422	KM199510
*P. rhizophorae*	MFLUCC 17-0416 ^T^	*Rhizophora apiculata*	Thailand	MK764283	MK764349	MK764327
*P. rhododendri*	IFRDCC 2399 ^T^	*Rhododendron sinogrande*	China	KC537804	KC537818	KC537811
*P. rhodomyrtus*	CFCC 55052	*Cyclobalanopsis augustinii*	China	OM746311	OM839984	OM840083
*P. rosarioides*	CGMCC 3.23549 ^T^	*Rhododendron decorum*	China	OP082430	OP185513	OP185520
*P. rosea*	MFLUCC 12-0258 ^T^	*Pinus* sp.	China	JX399005	JX399036	JX399069
*P. scoparia*	CBS 176.25 ^T^	*Chamaecyparis* sp.	China	KM199330	KM199393	KM199478
*P. sequoiae*	MFLUCC 13-0399 ^T^	*Sequoia sempervirens*	Italy	KX572339	-	-
*P. shaanxiensis*	CFCC 54958 ^T^	*Quercus variabilis*	China	ON007026	ON005054	ON005043
*P. shorea*	MFLUCC 12-0314 ^T^	*Shorea obtusa*	Thailand	KJ503811	KJ503814	KJ503817
*P. sichuangensis*	CGMCC 3.18244 ^T^	*Camellia sinensis*	China	KX146689	KX146807	KX146748
*P. silvicola*	CFCC 55296 ^T^	*Cyclobalanopsis kerrii*	China	ON007032	ON005060	ON005049
*P. spatholobi*	SAUCC231201 ^T^	*Spatholobus suberectus*	China	OQ692023	OQ718798	OQ718740
*P. spathulata*	CBS 356.86 ^T^	*Gevuina avellana*	Chile	KM199338	KM199423	KM199513
*P. spathuliappendiculata*	CBS 144035 ^T^	*Phoenix canariensis*	Australia	MH554172	MH554845	MH554607
*P. suae*	CGMCC3.23546 ^T^	*Rhododendron delavayi*	China	OP082428	OP185521	OP185514
*P. telopeae*	CBS 114161 ^T^	*Telopea* sp.	Australia	KM199296	KM199403	KM199500
*P. terricola*	CBS 141.69 ^T^	Soil	Pacific Islands	MH554004	MH554680	MH554438
*P. thailandica*	MFLUCC 17-1616 ^T^	*Rhizophora apiculata*	Thailand	MK764285	MK764351	MK764329
*P. trachycarpicola*	IFRDCC 2240 ^T^	*Trachycarpus fortunei*	China	JQ845947	JQ845945	JQ845946
*P. tumida*	CFCC 55158 ^T^	*Rosa chinensis*	China	OK560610	OL814524	OM158174
*P. unicolor*	MFLUCC 12-0276 ^T^	*Rhododendron* sp.	China	JX398999	JX399030	-
*P. verruculosa*	MFLUCC 12-0274 ^T^	*Rhododendron* sp.	China	JX398996	-	JX399061
*P. vismiae*	HHL-DG	*Rhizophora stylosa*	China	HM535704	HM573246	-
*P. yanglingensis*	LC4553 ^T^	*Camellia sinensis*	China	KX895012	KX895345	KX895231
*P. yunnanensis*	HMAS 96359 ^T^	*Podocarpus macrophyllus*	China	AY373375	-	-
*Neopestalotiopsis protearum*	CBS 114178 ^T^	*Leucospermum cuneiforme*	Zimbabwe	JN712498	KM199463	LT853201

^a^ Strains isolated from the current study are given in bold. ^T^ = ex-type culture.^b^ CFCC = China Forestry Culture Collection Center, China; ICMP = International Collection of Microorganisms from Plants, Auckland, New Zealand; LC = working collection of Lei Cai, housed at the Institute of Microbiology, Chinese Academy of Sciences, Beijing, China; IFRDCC = International Fungal Research and Development Culture Collection, Kunming, Yunnan China; CGMCC = China General Microbiological Culture Collection Center, Beijing, China; CBS = culture collection of the Westerdijk Fungal Biodiversity Institute, Utrecht, The Netherlands; MFLUCC = Mae Fah Luang University Culture Collection, Chiang Rai, Thailand; CSUFTCC = Central South University of Forestry and Technology Culture Collection, Hunan, China; BRIP = Plant Pathology Herbarium, Department of Employment, Economic, Development and Innovation, Queensland, Australia; MFLU = Mae Fah Luang University Herbarium, Thailand; HGUP = Plant Pathology Herbarium of Guizhou University, Guizhou, China; HMJAU = Herbarium of Mycology of Jilin Agricultural University, Jilin, China; SAUCC = Shandong Agricultural University Culture Collection, Taian, Shandong, China; NTUCC = The Department of Plant Pathology and Microbiology, National Taiwan University Culture Collection, Taipei, Taiwan (ROC); NOF = The Fungus Culture Collection of the Northern Forestry Centre, Alberta, Canada; E = The “Coleção de culturas de fungos fitopatogênicos Prof. Maria Menezes”, Universidade Federal Rural de Pernambuco, Recife, Brazil; CAA = culture collection of Artur Alves, housed at Department of Biology, University of Aveiro, Aveiro, Portugal; KNU = Kyungpook National University, Daegu, South Korea; NCYUCC = The National Chiayi University Culture Collection, Jiayi, Taiwan; ZHKUCC = the culture collection of Zhongkai University of Agriculture and Engineering, Guangzhou City, Guangdong, China; TAP = Tamagawa University, Tokyo, Japan; MAFF = Ministry of Agriculture, Forestry and Fisheries, Tsukuba, Ibaraki, Japan; MEAN = Instituto Nacional de Investigação Agrária e Veterinária I. P.; KUMCC = Kunming Institute of Botany Culture Collection, Yunnan, China; HMAS = Mycological Herbarium, Institute of Microbiology, Chinese Academy of Sciences, Beijing, China. ^c^ ITS = internal transcribed spacer; *TUB2* = b-tubulin; *TEF1* = translation elongation factor1-α.

## Data Availability

Data are contained within the article.

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
