# Peer review of "Pestalotiopsis jiangsuensis sp. nov. Causing Needle Blight on Pinus massoniana in China"

_jof, 2024, doi:10.3390/jof10030230_

Round 1
Reviewer 1 Report
Pestalotiopsis jiangsuensis sp. nov. was here described as a new species in the genus Pestalotiopsis that has been identified as a causal agent of needle blight on Pinus massoniana (known as Masson's pine or Chinese red pine). Its occurrence has been reported particularly in Jiangsu province so far, but the understanding of its distribution and possible transmission to other regions is crucial for managing and mitigating its impact on pine forests. Continued research is needed to further elucidate the biology, epidemiology, and genetic diversity of P. jiangsuensis, to develop sustainable and integrated approaches for managing needle blight disease in pine forests. The present article gives significant input into this knowledge, highlighting the importance of ongoing research and management efforts in forest pathology and plant protection.
This is the first report of the occurrence of this pathogen on Chinese red pine, so it is worth mentioning it in the title or the abstract.
The abstract lacks results: it was only mentioned that “The study revealed the diversity of pathogenic species of needle blight on Pi. massoniana” without any details.
The Keywords clearly reflect the paper's content.
The introduction presents the problem, although needs some precision in line 62, in which “different geographical area” pine needle blight was identified, or give a citation.
Experimental methods concerning the DNA analysis and phylogenetic study are adequate.
Results and Discussion need improvements:
- in Table 2, except those studied (CFCC 59538-42), some abbreviations of strains (e.g. SS1-033I, SY17A, HHL-DG) are without any explanation of their origin
- lines 208-246: information here is given without any text phrasing – if such a form is accepted by the Journal editor, I do not complain. Generally, such data is crucial for the first report about the occurrence of one species in a new host.
- Lines 257-259: the reference to the morphology of Pestalotiopsis funereal is not supported by any graph or citation.
- Line 155: from the dendrogram Fig. 2, the highest sequence similarity between 5 investigated barcoding based on multiple loci is and P. foliicola, P. pinicola, P. suae, and P. rosea is ML/BI = 95 /0.99, and not 98/0.92. Please confirm
References are complete and adequate but their citation in the text need correction:
- line 22 and following: I suggest removing an extra “1” from all citations, starting from [1.1.], where [1] is sufficient. Otherwise all citations, like e.g. “[1.221.251.26]” in line 51 is completely incomprehensible
The length of the manuscript is commensurate with the paper's content.
I recommend publishing the article after those corrections.
Reviewer 2 Report
The paper “Pestalotiopsis jiangsuensis sp. nov. causing needle blight on Pinus massoniana in China” by Li et al. is interesting and it deserves publication after some revision, in my opinion.
The paper is well written but needs revision. First of all, the references are wrongly indicated along the all text. They are not well referred. Please correct this. This makes the paper hard to follow in terms of the literature that is cited.
I found that there are no conclusions. I think this is a weakness. In my opinion, this must be fixed!
The list of references is satisfactory but could be enlarged, in my opinion, in terms of biogeography and epistemology of the diseases caused by Pestalotiopsis spp. Please, correct references along the text. They are wrongly indicated as I said before.
Comments:
The paper “Pestalotiopsis jiangsuensis sp. nov. causing needle blight on Pinus massoniana in China” by Li et al. is interesting and it deserves publication after some revision, in my opinion.
The paper is well written but needs revision. First of all, the references are wrongly indicated along the all text. They are not well referred. Please correct this. This makes the paper hard to follow in terms of the literature that is cited.
The tittle is well chosen.
The Abstract is clear, concise, pointing out the main results. Correct “were” by “was” as it is singular (line 15). “A new species was described”… But the abstract could be more developed.
Keywords: I would not repeat “Pestalotiopsis” in the keywords as it is already in the tittle.
The Introduction is well written, and the objective of the study is clearly stated; One thing: there are some species where the author´s name for that species is given (this must be only at the first time when the given species is referred), but others do not have the author’s name. Please uniform this situation!
Materials and Methods could be more well explained and described, at least in section 2.1
In section 2.1: please explain why you sampled only 5 trees (only those were symptomatic?). Please refer how many isolates you gathered! Please explain why you got only five isolates, if you got only 5 isolates. This is not clear to me! And is one isolate of each one of one of the five trees? This is not clear to me once more. It would be nice a scheme or a table, perhaps.
We got only some of this information in the results subsection 3.1
Nevertheless, the techniques and methodologies were well chosen.
Line 69. Correct m2 to m2.
Results are well described and illustrated by useful tables and figures. Photos of the morphology of the fungus are of great quality.
In the tittle of Table 2, please correct “anlysesa” to” analyses”.
Tittle of section 3.3 It is “Taxonomy” not “Taxonomys”. Taxonomy in plural is not used and it should be “taxonomies”. Please, be sure that the name of the species is already valid before publication and that is has been curated. I assume it is!
SubSection 3.4 Pathogenecity test:
Line 261. Please insert “The” before “three representatives”.
Line 264. Please correct “Massoniana” to “massoniana”.
Lines 266 and 267. Please explain and rephrase “In this study, the pathogenicity of Pestalotiopsis jiangsuensis is strong, which may relate to its high isolation rate.” It is not clear to readers in my opinion. For example “In our test, Pestalotiopsis jiangsuensis revealed to be a strong pathogen as …, and this may be related to its high isolation rate (please, explain why!)
Did the authors check by sequencing the identity of the isolates that were re-isolated from 100% of the inoculated plants? I believe so, but this could be said at the end of this section (in lines 267 and 268), before saying that the Koch’s postulates had been fulfilled.
Discussion of the results is relatively concise and well organised. But in my opinion, it could be more developed, comparing for example the degree of severity and pathogenicity of some of the Pestalotiopsis spp. in Pinus spp. And even in other hosts. Also, some explanation about the arise of this new species could be given or at least discussed. Could it be caused by host transfer or derived form a common ancestor of another Pestalotiopsis species? This could be explored in the literature to enrich the discussion, if the authors do find this important.
The last paragraph of the discussion could be more developed. This is the most important part of the Discussion for me. For example, try to discuss about the diversity of the Pestalotiopsis species. Line 315: see here …”which indicated that the pathogens of the same genus on the same host were diverse”… Please try to explain better and develop more!
Line 321: Please put “Pinus” in italics.
I found that there are no conclusions. I think this is a weakness. In my opinion, it would be important to state the main conclusions of the article!
The list of references is satisfactory but could be enlarged, in my opinion, in terms of biogeography and epistemology of the diseases caused by Pestalotiopsis spp. Please, correct references along the text. They are wrongly indicated as I said before.
Round 2
Reviewer 1 Report
After a careful revision of the author’s response to the first review and the latest version of the manuscript, I think that the work has been extensively improved
I have no detailed comments.
Reviewer 2 Report
The paper was much improved and all my issues were properly adressed by the authors. I have no further questions.
As said above, all my doubts and questions were properly answered and the Discussion was improved as requested. Also, Conclusions were added!